# Numerical Computation of Anisotropic Thermal Conductivity in Injection Molded Polymer Heat Sink Filled with Graphite Flakes

**DOI:** 10.3390/polym14163284

**Published:** 2022-08-12

**Authors:** Robert Brachna, Jan Kominek, Michal Guzej, Petr Kotrbacek, Martin Zachar

**Affiliations:** Heat Transfer and Fluid Flow Laboratory, Faculty of Mechanical Engineering, Brno University of Technology (BUT), Technicka 2896, 616 69 Brno, Czech Republic

**Keywords:** polymer heat sink, graphite flakes, anisotropy, thermal conductivity, heat conduction, numerical simulation

## Abstract

The use of polymer composites as a replacement for commonly applied materials in industry has been on the rise in recent decades. Along with the development of computer software, the desire to predict the behavior of new products is thus increasing. Traditional additives in the form of fibers cause anisotropic properties of the whole product. The subject of the presented study is a polymer heat sink prototype with a thermally conductive filler in the form of graphite flakes, which differs from the commonly used fibers. Three simplified approaches are introduced to model the thermal conductivity anisotropy of an entire heat sink. Each model is subjected to an inverse heat conduction problem, the output of which are thermal conductivity values. These are optimized to minimize the difference between simulated and experimental temperatures at selected locations in the model. The approaches are compared with each other with respect to their error against the experimentally obtained results. The goal is to find a sufficiently simplified approach that can be applied to products of various geometries. This would remove the costly and time-consuming need for mold production and experimental testing.

## 1. Introduction

In thermal management applications, the greatest requirement is to maintain a stable temperature below the maximum operating temperature specified by a manufacturer. Demands in this area are constantly increasing. The sizes of electronic components have shrunk significantly over the past decades, while their thermal performance has increased rapidly with advances [1,2]. Typical materials used for heat sinks in the electronics industry are aluminum or copper [2,3,4,5]. They owe this prominence primarily to their high thermal conductivity.

Over the last century, polymer materials have been used extensively in the industrial world. They are well known as thermal insulators, but despite their low thermal conductivity, they have found their way into thermal management applications such as heat exchangers [6,7,8,9]. By combining polymers with different fillers, they can not only counter conventional materials in their respective fields, but even successfully replace them. By using a thermally conductive filler, the thermal conductivity of the resulting composite can be increased many times over. In this respect, the most attention is paid to graphite because of its high thermal conductivity [10,11]. Heat sinks made of thermally conductive polymer composites bring competition to conventional heat sinks in the form of lower weight and less energy intensive production. Their other potential advantage is shape flexibility. They can be easily made into a variety of geometries.

The resulting thermal conductivity of a polymer composite is a combination of the thermal conductivity of both polymer and filler. The research regarding carbon-based fillers can be divided into two categories based on the dimensionality of the filler used. Carbon nanotubes and carbon fibers are one-dimensional, having a high thermal conductivity along the longitudinal direction [12]. The primal focus is on utilizing the ultra-high single axis thermal conductivity and thus enhancing the overall thermal performance of the composite [13,14]. Graphite flakes are considered as two-dimensional fillers. The in-plane thermal conductivity of graphene—the building block of graphite, has been reported to be as high as 5300 W/m·K [15]. Therefore, graphite poses a great potential as a thermal conductivity enhancer [16]. However, its through-plane thermal conductivity is an order of magnitude lower due to weaker chemical bonds between graphene layers [17]. This creates an anisotropic behavior of the composite based on the internal filler orientation. Part of the research on graphite flake polymer composites is focused on thermal interface materials [18,19]. Related to this is the desire to influence the resulting orientation of the flakes in order to make appropriate use of their high intrinsic thermal conductivity [19,20,21].

Although 3D printing of polymer composites is slowly becoming a favorite production solution [21,22], traditional injection molding is still dominant. Fiber-based composites exhibit a shell-core structure that has been documented and numerically verified many times [23,24,25,26]. This morphology has several variations, which depends on the thickness of the sample produced. Typically, it involves a heterogeneity in fiber orientation through the part thickness. In the core, they exhibit an orientation perpendicular to the flow velocity vector, whereas conversely in the shell they are oriented parallel to this vector [23]. In the case of graphite flakes as fillers, the morphology description is not unified. Many literature sources state that the graphene planes are oriented in the main melt flow direction and thus that in-plane conductivity is higher than through-plane conductivity [21,27,28]. No mention is made of possible heterogeneity of morphology across the thickness of the sample. However, in the study conducted by Grundler et al. [29], a similar pattern to that of the carbon fiber molds was observed, where a shell-core structure was formed. This was additionally supported by thermal conductivity measurements in different configurations. The same shell-core internal structure is exhibited by the heat sink prototype investigated in this paper.

Composite models can be divided into analytical models and finite element simulations. The use of analytical methods is limited to specific cases and thus in complex geometries, such as a heat sink, a numerical approach is desirable [30]. Work addressing the numerical determination of the anisotropic thermal conductivity of a complex product in relation to experimental data is scarce in the literature. Of note are the studies by Czajkowski et al. [31] and Chanfa et al. [32], where the task of determining thermal conductivities is formulated as an inverse heat conduction problem.

In this study, a prototype of a graphite flake-based polymer heat sink is used as a basis for the development of numerical models. Three different qualitative assumptions are made on the thermal conductivity distribution in the heat sink geometry. The first one does not consider the anisotropy caused by graphite flakes. The second model is based on the most common results reported in the literature, namely the distinction of thermal conductivity into through-plane and in-plane. The third, the most detailed, uses information about the internal orientation of the graphite planes in the heat sink geometry. On this basis, a shell-core model is developed in which the dimensions of the relevant layers are estimated from microscopic images. All three models are parameterized using the principal thermal conductivities. The values of these parameters are computed by an inverse optimization task whose inputs are experimentally obtained temperatures.

## 2. Materials and Methods

### 2.1. Heat Sink Prototype

The prototype’s base had dimensions of 106 × 58 × 10 mm, containing 9 fins with dimensions of 6 × 30 × 50 mm and an offset of 11 mm. The material used for production is composite, technically designated as TT-6600-5001 EC, provided by Avient. The base matrix is thermoplastic polyamide 66, which is enriched with a thermally conductive filler in the form of graphite flakes. According to the official datasheet provided by the manufacturer, the in-plane thermal conductivity ranges from 19 to 21 W/m·K and the through-plane from 4.5 to 5.5 W/m·K (other basic properties: density—1.64 g/cm^3^, tensile modulus—14.7 GPa). The heat sink was manufactured by injection molding. The mentioned material was melted and injected into a mold that was designed specifically for this research (see Figure 1). Several variations of heat sinks were produced based on the melt inlet side of the heat sink.

The influence of melt entrance positioning on the overall heat sink’s thermal behavior was studied in [33]. The research led to the conclusion that the entrance along the longer base’s edge leads to a better thermal efficiency. However, the shorter edge was chosen for this study since the internal material structure of the heat sink is easier to properly account for in the numerical model.

### 2.2. Measurement Description

The heat sink was subjected to experimental measurements, which were well described in studies [33,34]. It is reasonable to mention the aspects that are important for a proper numerical model. The placement of the heat sink (according to Figure 2) permitted positioning the model freely in space. The thermostatic chamber ensured a stable air temperature of 20 °C. The heater on the heat sink’s base provided thermal power output of 15 W. To ensure a good thermal contact, thermal paste was applied on the heat sink—heater surface interface.

The output data of the experiment consists of the temperature from the thermocouple attached to the heater’s body, the average temperatures at the top of the fins obtained from the thermal imaging camera (FLIR E5), and the temperatures from another 6 thermocouples located inside the heat sink as shown in Figure 3.

### 2.3. Material Study

In the research conducted by Grundler et al. [29], the base matrix of the composite was made up of polyamide 6, making it an identical material in type. In their research, they used a 2 mm thick test specimen produced by injection molding. Microscope images of the fractured specimen show the orientation of the graphite flakes (see Figure 4). The structure can be imaginatively divided into a core and a shell.

The flakes in the **core** are arranged in a **parabolic shape** perpendicular to the direction of material flow during fabrication.Conversely, in the **shell** they form **dense parallel lines** in the flow direction.

The core was 1 mm thick and the shell 0.5 mm thick. We introduce the terms good and poor conductivity in connection with the orientation of the flakes:Good—parallel to the flakes’ orientation,Poor—perpendicular to the flakes’ orientation.

**Figure 4 polymers-14-03284-f004:**
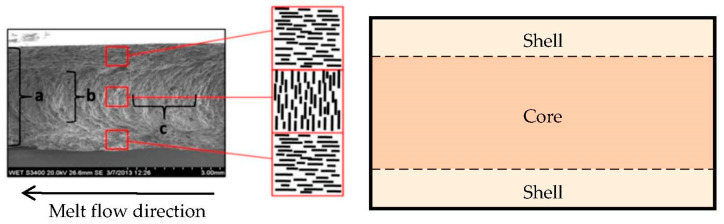
The internal structure of the test specimen in the study conducted by Grundler et al. [29].

In the next part of the research, they measured the thermal conductivity using the laser flash method. The measurement was performed in three different configurations:whole specimen, perpendicular to the flow (in the Figure 4 depicted as (**a**)—overall through-plane conductivity,ground specimen (the shells were ground, so that the core itself could be measured), perpendicular to the flow (**b**)—through-plane conductivity in the core,ground specimen, parallel to the flow (**c**)—in-plane conductivity in the core.

The values of thermal conductivities as a function of temperature are given in Figure 5. The conductivities in the shell were not investigated in this study. It is reasonable to assume that these are different from those in the core because

flakes in the core form a parabolic structure,flakes in the shell are structured into dense parallel lines.

Since these different layers are analyzed from a macroscopic perspective, their different internal nature of the flake arrangement should lead to different good and poor thermal conductivities.

**Figure 5 polymers-14-03284-f005:**
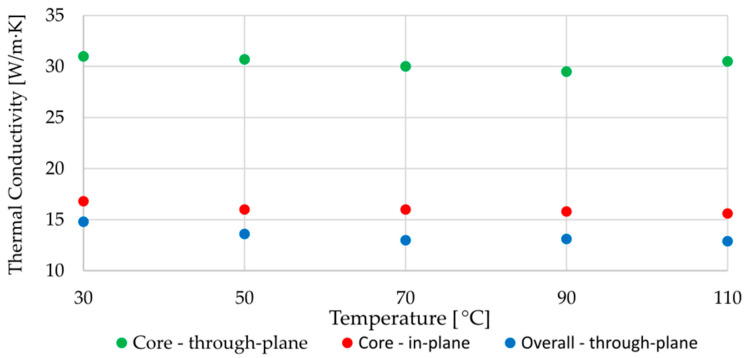
Measured thermal conductivities in the study by Grundler et al. [29].

However, using the available data, the poor conductivity in the shell can be calculated/estimated (will be referred to as **d**) using series connected thermal resistances. The procedure is analogous to the resistors in electrical circuits and is well described in [35]. The scheme for the calculation is shown in Figure 6.

The resulting resistance is given as the sum of the sub-resistances:(1)$$R_{a}=R_{b}+2R_{d}.$$

The only unknown resistance $R_{d}$ can be obtained using the following relations:(2)$$R_{d}=\frac{R_{a}-R_{b}}{2},\;\;R_{a}=\frac{L_{a}}{\lambda_{a}},\;\;R_{b}=\frac{L_{b}}{\lambda_{b}}.$$

Then the estimated value of poor conductivity in the shell follows as
(3)$$\lambda_{d}=\frac{L_{d}}{R_{d}}.$$

Applying this set of equations to the conductivity data at 70 °C (from Figure 5), we arrive at the value $\lambda_{d}=8.3$ W/m·K, which is approximately two times lower than the poor conductivity in the core. This rough estimate (relative ratio between poor conductivities in the core and shell) is later used in one of the numerical models.

A fractural analysis was performed on the entire heat sink prototype. Figure 7 shows the internal structure of a fractured heat sink’s fin, which can be considered as a thin test specimen as in the study by Grundler et al. Based on the similarity of the structure, the conclusions drawn from the study are applied in modelling of the heat sink, with the core and shell sizes determined from the fractural analysis.

The region where the flakes’ orientation transitions from a parabolic structure to parallel lines is considered as the core-shell interface (represented by the red dashed line in Figure 7). The core and shell dimensions were obtained from a graphics software after manually specifying the core-shell interface. The values used to create the numerical model of the composite heat sink are averaged from several photographs. The dimensions were determined separately for the heat sink’s base and the fins.

### 2.4. Computational Method

Computational problems in heat conduction are traditionally divided into forward and inverse tasks. The determination of material properties such as thermal conductivity is an inverse task. In the course of the calculation, the associated forward problem has to be repeatedly solved due to the iterative nature of inverse formulations. The forward task is described by a differential heat conduction equation. We are interested in its steady state form
(4)$$-\nabla^{T}\left(K\nabla T\right)=Q.$$

For solvability, boundary conditions need to be added to this equation, which are given in the form of a prescribed temperature (Equation (5)) or a known heat flux at the surface (Equation (6)):(5)$$T=T_{s},$$
(6)$$-n^{T}K\nabla T=\dot{q}.$$

The subject of interest is the thermal conductivity tensor K, which in general contains 9 unknown components. However, it is proved that the conductivity tensor is symmetric and positively definite [36]. Consequently, there exists a coordinate system in which the conductivity is of a diagonal form
(7)$$K=\left(\begin{array}{ccc} \lambda_{1} & 0 & 0\\ 0 & \lambda_{2} & 0\\ 0 & 0 & \lambda_{3} \end{array}\right).$$

The problem of describing the conductivity tensor is thus reduced to determining the values of the principal conductivities $\lambda_{1},\lambda_{2},\lambda_{3}$ and the corresponding coordinate directions.

Comsol Multiphysics is used as a forward task solver. Three models corresponding to the heat sink experiment are created. They differ in the assumptions on the qualitative distribution of the thermal conductivity, which are explained in later sections. The inverse task is formulated as an optimization problem, where the values of the parameterized conductivities of a given model are sought to minimize the difference of the measured temperatures from the experiment with the corresponding temperatures obtained from the numerical model. We include the parameterized conductivities in the parameter vector λ, and then the optimal values of $\lambda^{*}$ are the solution to the problem
(8)$$\lambda^{*}=\arg\min\left\{f\left(\lambda \right)\right\}.$$

To evaluate the objective function f, the forward task is first solved using the conductivities λ and then the sum of the squared errors of the corresponding temperatures is calculated
(9)$$f\left(\lambda \right)=\overset{N_{T}}{\underset{i=1}{\sum }}w_{i}{\left({\tilde{T}}_{i}-{\bar{T}}_{i}\;\right)}^{2}.$$

The weights are chosen so that the total weights from the temperatures at the fins, the thermocouples in the heat sink, and the thermocouple near the heat source are equal. The weights within each temperature category are the same. With a requirement for the total sum of the weights to be equal to one, we have
(10)$$\underset{\mathrm{fins}}{\sum }w_{i}=\underset{\begin{array}{c} \mathrm{heat}\\ \mathrm{sink} \end{array}}{\sum }w_{i}=w_{\mathrm{source}}=\frac{1}{3}.$$

The Nelder–Mead simplex method is used as the optimization algorithm, which is implemented in Matlab for the purpose of this research. Using the available tool Comsol LiveLink for Matlab, these softwares are linked together.

## 3. Numerical Model

The geometry of the created model consists of the heat sink and the heater on the heat sink’s base. Internally, the geometry is divided into subdomains in order to model the internal structure (Figure 8).

From the material properties, only the thermal conductivity is necessary for stationary heat conduction. In the case of the aluminum body, it is set to 130 W/m·K (conductivity of the used aluminum alloy) and in the active heater domain, 400 W/m·K is specified so that no significant temperature gradient is created and thus it serves solely as a heat source. Corresponding to the experiment, the total power of the domain is set to 15 W. Material description of the heat sink is covered in the section “Approaches to Thermal Conductivity”.

The model does not contain a fluid flow domain; the interaction of the surface with buoyancy-driven air flow is accounted for by using natural convection as a boundary condition. The other boundary conditions are radiation and thermal contact at the heat sink—heater interface. Natural convection is modelled by the heat transfer coefficient; the corresponding heat flux for convection appearing in Equation (6) is then of the form
(11)$$\dot{q}=h\left(T-T_{ext}\right).$$

In order to use the most accurate heat transfer coefficient and at the same time not to increase computational complexity of the numerical model, a separate CFD simulation with properties corresponding to an aluminum heat sink was created. From this simulation, heat transfer coefficient values were obtained as a function of the heat sink surface temperature (related to the ambient air temperature of 20 °C corresponding to the regulated air conditions during the measurement). These values were imported into Comsol as a temperature dependent interpolation function. The radiation is divided into emission to the surrounding environment (the walls of the measuring box) and to the interaction between the fins’ surfaces (Figure 9).

In the case of ambient radiation, the heat flux is calculated using the specified surface emissivity according to
(12)$$\dot{q}=\sigma \varepsilon \left(T^{4}-T_{ext}^{4}\right).$$

For interacting surfaces, the heat flux formula is in a more general form as the difference of the outgoing (radiosity) and incoming (irradiation) radiant energy
(13)$$\dot{q}=J-G.$$

The description of the calculation of these quantities is technically demanding; together with the necessary derivation it can be found in the book [35].

The boundary condition of thermal contact accounts for the effect of the applied thermal paste. The heat flux along the normal direction for the separated surfaces (see Figure 10) is calculated jointly using the relations
(14)$${\dot{q}}_{1}=h_{c}\left(T_{2}-T_{1}\right),\;{\dot{q}}_{2}=h_{c}\left(T_{1}-T_{2}\right).$$

The value of contact conductance is modelled by a constant, which was calculated using a separate aluminum heat sink of the same geometry. The aluminum model was further used for overall verification and calibration of the numerical model due to the known thermal conductivity of aluminum.

The mesh (see Figure 11) was created by discretizing the xy-plane, which was subsequently swept in the z-direction. The emphasis was on using regular hexahedral elements, the exception being the radiuses of the fins, where a triangular mesh was used in the xy-plane. During the meshing process, a sensitivity analysis on the size (number) of computational elements was performed. The process started with a coarse mesh. At each refinement step (performed manually), the coarser mesh was compared with a finer one, observing temperature differences at the selected locations in the model (at those corresponding to the experimentally obtained temperatures). The process was done as soon as the differences between meshes dropped below 0.01 °C. The final mesh is composed of approximately 70,000 hexahedral elements and 4500 prism elements with the average element quality of 0.48.

All input parameters to the numerical model except thermal conductivity of the composite are summarized in Table 1.

## 4. Approaches to Thermal Conductivity

The output of the numerical model (after performing the proposed inverse formulation) is the thermal conductivity tensor K. This chapter describes its parameterization for three different assumptions.

### 4.1. Isotropy

The whole heat sink is considered as isotropic. The thermal conductivity tensor is described by a single parameter that is the same throughout the heat sink’s geometry, that means
(15)$$K=\left(\begin{array}{ccc} \lambda & 0 & 0\\ 0 & \lambda & 0\\ 0 & 0 & \lambda \end{array}\right).$$

### 4.2. Simple Anisotropy

By isolating one principal conductivity, a simple anisotropic model is created. In connection with the different direction of melt flow in the base and in the fins, the conductivities in the xy-plane are interchanged, in the following sense:(16)$$K_{\mathrm{base}}=\left(\begin{array}{ccc} \lambda_{1} & 0 & 0\\ 0 & \lambda_{2} & 0\\ 0 & 0 & \lambda_{1} \end{array}\right),\;\;K_{\mathrm{fin}}=\left(\begin{array}{ccc} \lambda_{2} & 0 & 0\\ 0 & \lambda_{1} & 0\\ 0 & 0 & \lambda_{1} \end{array}\right).$$

By this choice, the parameter $\lambda_{1}$ is interpreted as the in-plane thermal conductivity and $\lambda_{2}$ being the through-plane thermal conductivity.

### 4.3. Fractural Anisotropy

The third approach uses knowledge of the internal structure throughout the heat sink. The domains corresponding to the core of the polymer material, have the principal conductivities parameterized
(17)$$\lambda_{1}^{\mathrm{core}}=\lambda_{2}^{\mathrm{core}},\;\;\lambda_{3}^{\mathrm{core}}.$$

The above equality is based on the property of graphite. The shells have separate parameters of the principal conductivities
(18)$$\lambda_{1}^{\mathrm{shell}}=\lambda_{2}^{\mathrm{shell}},\;\;\lambda_{3}^{\mathrm{shell}}.$$

Using the conclusion drawn from the study by Grundler et al., the poor conductivities in the core and shell are bound by the relationship
(19)$$\lambda_{3}^{\mathrm{core}}=2\lambda_{3}^{\mathrm{shell}}.\;$$

The good conductivities are assumed to be equal, that is
(20)$$\lambda_{1}^{\mathrm{core}}=\lambda_{1}^{\mathrm{shell}}.\;$$

The associated coordinate directions are set according to the microscopic images obtained from the fracture analysis. In addition, a non-trivial rotation of the thermal conductivity tensor is applied in the corner sections of the geometry.

## 5. Results

The presented thermal conductivity models were subjected to the proposed inverse task, the output of which were the parameterized principal conductivities with corresponding temperature fields (see Figure 12). The resulting values of conductivities are shown in Table 2.

Although the values of the objective function (computed as stated in Equation (9)) do not at first glance show the true temperature errors of the simulation, the four times higher objective value of the isotropic model confirms that such a complex material should not be simplified to isotropy.

Fractural anisotropy exhibits the lowest error in terms of the objective function. A comparison of the thermal image from the experiment and the simulated temperature field (fractural model) is captured in Figure 13. However, to evaluate whether the increased complexity significantly improves the thermal behavior against the simple anisotropic approach, the temperature differences must be analyzed. All target temperatures are captured in Figure 14. The error of the isotropic model is clearly visible, its deviations are in most cases between 2 and 3 °C.

Considering the temperature at the heat source both anisotropic and fractural approach agreed with the measured value. Noticeable differences between the latter approaches are captured in the average temperatures on the top of the fins. For a better comparison, the errors against the experimentally measured values are shown in Figure 15. Around the middle fin, the fractural model is about 0.3 °C more accurate. In terms of squared differences, this is reflected in higher objective value of the anisotropic model. The same is true for the errors at the thermocouple locations. Overall, the improvement using the fractural model is only in fractions of °C.

However, the advantage of the fractural approach is in its connection to the real heat sink prototype. The final optimized principal conductivities reflect the previous studies and assumptions. The numerical model preferred high thermal conductivity in the plane of graphite flakes. Since the datasheet by the manufacturers only provides through-plane and in-plane thermal conductivities without any mention about a possible difference in the shell, a final conclusion about the optimized values is not drawn. However, the values are close to the provided ones by the manufacturer (good conductivity in the core 17.9 W/m·K vs. in-plane 19–21 W/m·K, poor conductivity in the core 4.2 W/m·K vs. through-plane 4.5–5.5 W/m·K). Concerning the anisotropic approach, the optimized parameters lack real physical interpretation. They are a combination of the behavior of the core and shell.

## 6. Conclusions

The use of polymer-based materials has increased dramatically since their invention. The composites being created can outperform the dominant material used in a specific industrial field. Highly thermally conductive composites can compete with conventional aluminum heat sinks. To assess the performance of the new product under development, it is advisable to perform proper numerical simulations. However, polymer composites often exhibit anisotropic properties due to the orientation of the filler used—graphite flakes in the case of the investigated heat sinks. In order to produce a high-fidelity numerical simulation, the anisotropy needs to be sufficiently captured.

In this paper, a prototype of the polymer heat sink has been presented along with its numerical model. Three models with different assumptions on its thermal conductivity have been developed:**isotropic**—whole geometry with a single conductivity parameter,**anisotropic**—traditional approach with in-plane and through-plane thermal conductivities,**fractural**—thermal conductivity distribution based on the internal structure observed from microscopic images of the fractured heat sink.

The parameters of each model were optimized to minimize the difference with the experimentally obtained temperatures. The first model assessing the potential suitability of modelling over an isotropic material proved to be inadequate. The remaining approaches incorporating thermal conductivity anisotropy showed similar temperature results, but qualitatively represent a different reality.

The use of the anisotropic model poses simplicity in the sense that it is not necessary to verify the sizes of the core and the shells of the material. However, it must be taken into account that the sufficiency of this model has only been shown in this particular example. In the case of making further interventions in the final product, such as milling and thus artificially shrinking the shell, this model may not be sufficient in terms of thermal performance.

The benefit of the fractural model is its transferability to other geometries, as the results are consistent with the expected reality. Although its error compared to the experiment reached above 1 °C in certain locations, this is an acceptable shortcoming given the complexity of the problem being modeled. In contrast, the simple isotropic model exhibited errors up to 4 °C. Thus, with knowledge of the core and shell dimensions, the fractural approach can be used to perform reliable numerical simulations prior to actual product fabrication.

## Figures and Tables

**Figure 1 polymers-14-03284-f001:**
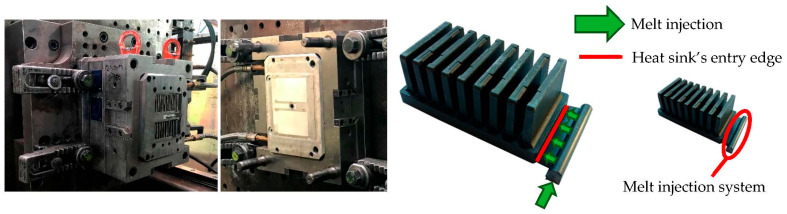
Both parts of mold specifically designed for this research (**left**). Entrance of the melt during the manufacturing process (**right**). The injection system was removed afterwards.

**Figure 2 polymers-14-03284-f002:**
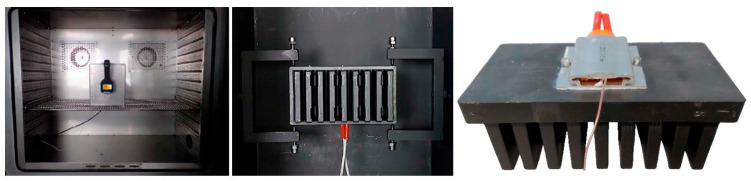
Measurement box inside the thermostatic chamber with attached thermal imaging camera (**left**), heat sink inside the measurement box (**middle**), attached heater with the thermocouple on the heat sink’s base (**right**).

**Figure 3 polymers-14-03284-f003:**
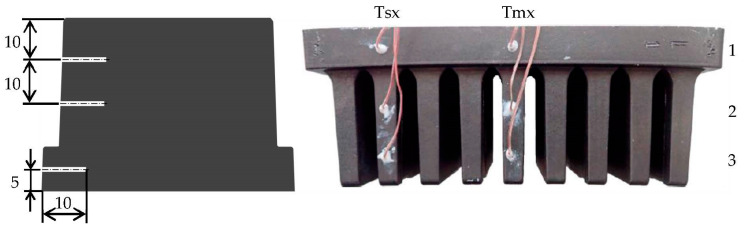
Locations of the thermocouples inside the heat sink. The locations in the middle fin will be referred to as Tmx, x being the ordinal number from the base up to the fin, e.g., Tm1—thermocouple in the base under the middle fin. Similarly, the locations of thermocouples in the outer fin will be labelled as Tsx. Dimensions are in mm.

**Figure 6 polymers-14-03284-f006:**
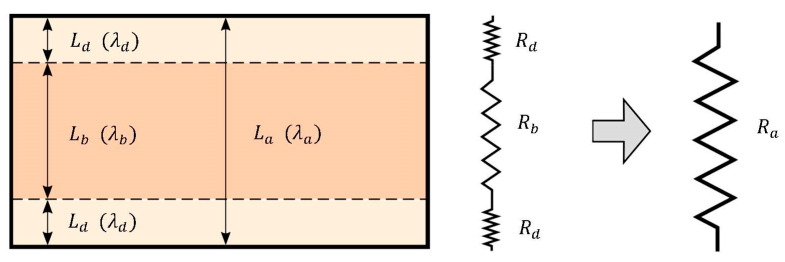
Calculation scheme for estimating the poor thermal conductivity in the shell.

**Figure 7 polymers-14-03284-f007:**
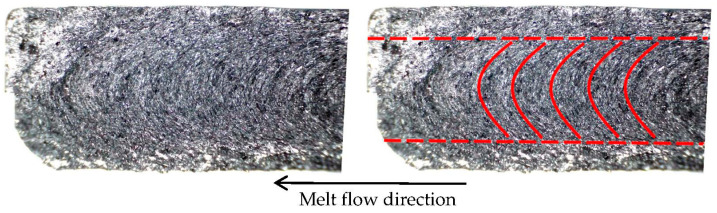
Internal structure along the height of a fractured fin of the heat sink prototype (**left**) with highlighted flakes’ orientation (**right**).

**Figure 8 polymers-14-03284-f008:**
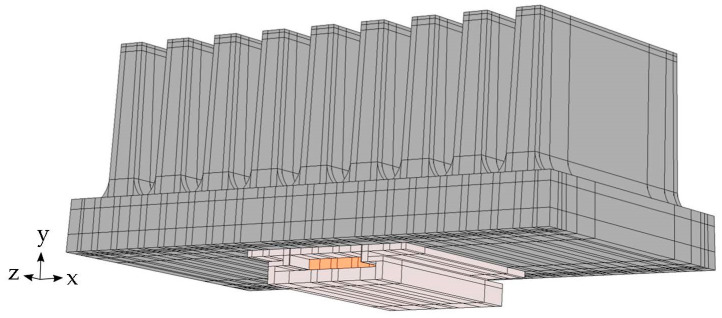
Geometry of the numerical model. Dark grey—heat sink, light grey—aluminum body of the heater, orange—active heater domain.

**Figure 9 polymers-14-03284-f009:**
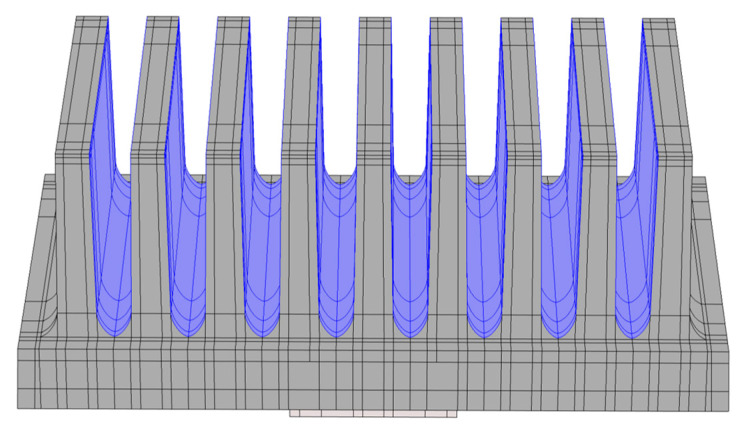
Fins’ surfaces (highlighted in blue) that are treated within the mutual radiation.

**Figure 10 polymers-14-03284-f010:**
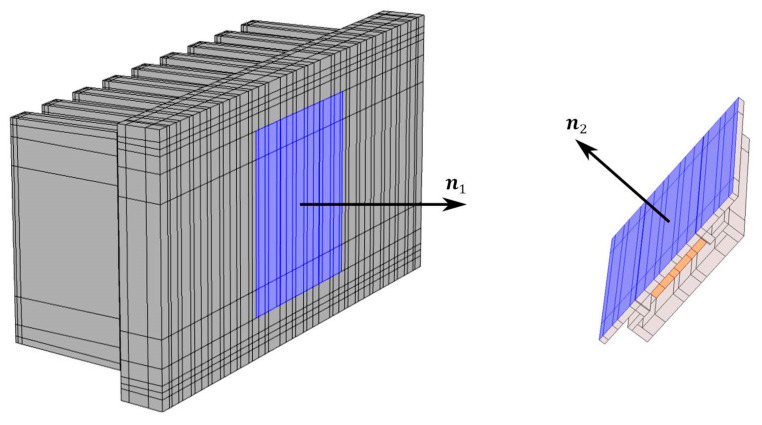
Thermal contact applied on the contact surfaces between the heat sink and the heater.

**Figure 11 polymers-14-03284-f011:**
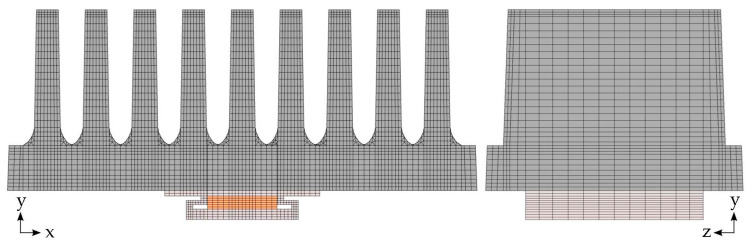
The mesh of the numerical model, view on the xy-plane (**left**) and on the yz-plane (**right**).

**Figure 12 polymers-14-03284-f012:**
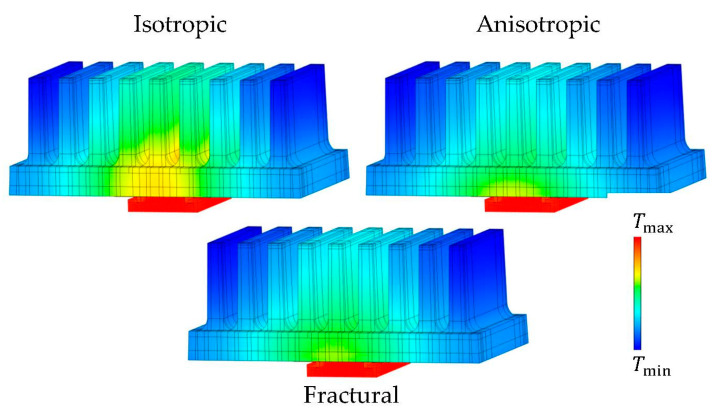
Temperature fields of optimized numerical models. Isotropic approach (**top left**)—$T_{\min}=43\;{}^{\circ}C$, $T_{\max}=77.3\;{}^{\circ}C$, anisotropic (**top right**)—$T_{\min}=45.6\;{}^{\circ}C$, $T_{\max}=78.6\;{}^{\circ}C$, fractural (**below**)—$T_{\min}=45.3\;{}^{\circ}C$, $T_{\max}=78.5\;{}^{\circ}C$.

**Figure 13 polymers-14-03284-f013:**
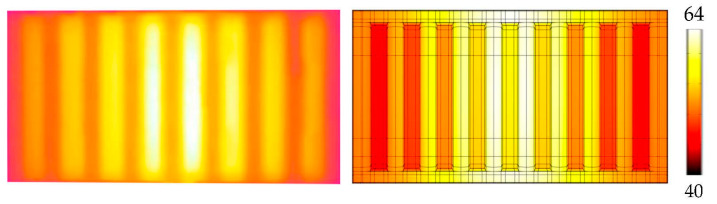
Thermal picture from the experiment and its comparison with the simulated temperature field (fractural model). The values by the color scale are in °C.

**Figure 14 polymers-14-03284-f014:**
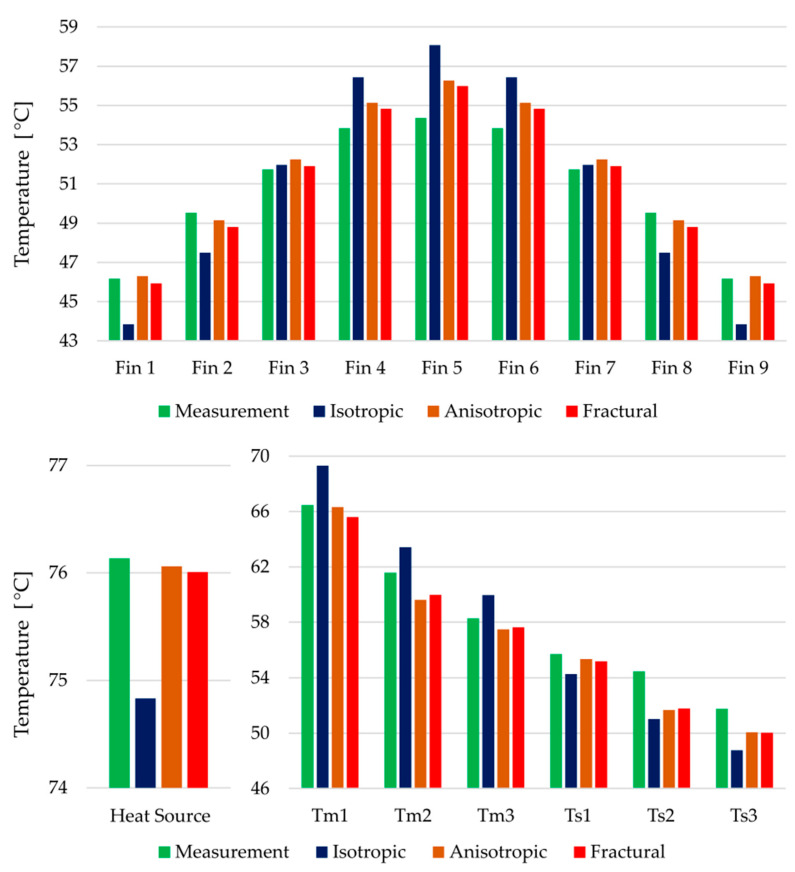
Measured and simulated temperatures of the three thermal conductivity approaches. Averaged temperatures on the top of the fins (**above**), temperature near the heat source (**below left**), thermocouples inside the heat sink (**below right**)—their labels correspond to the Figure 3.

**Figure 15 polymers-14-03284-f015:**
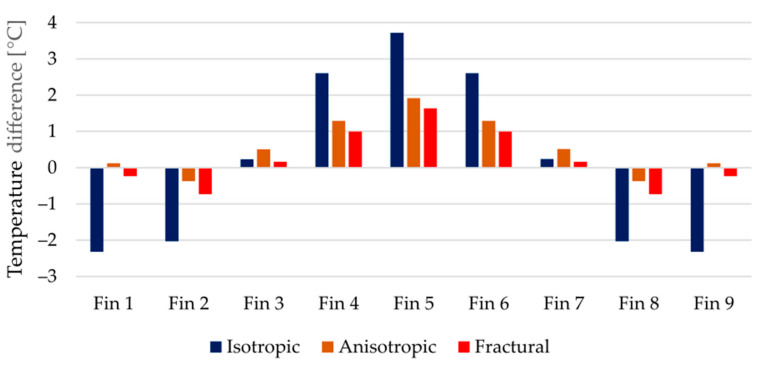
Differences between measured and simulated averaged temperatures on the top of the fins.

**Table 1 polymers-14-03284-t001:** Input parameters of the numerical model.

Parameter	Value	Description
** $\lambda_{Al}$ **	130 W/m·K	Thermal conductivity of the aluminum domain
** $\lambda_{heat}$ **	400 W/m·K	Thermal conductivity of the heater domain
** P **	15 W	Thermal power of the heater domain
** $T_{ext}$ **	20 °C	Surrounding temperature (ambient)
** $\varepsilon_{Al}$ **	0.1	Surface emissivity of the aluminum domain
** $\varepsilon_{HS}$ **	0.85	Surface emissivity of the composite heat sink
** $h_{c}$ **	7700 W/m^2^·K	Contact conductance at the interface between heat sink and the heater’s aluminum body

**Table 2 polymers-14-03284-t002:** Results of the inverse task for the presented thermal conductivity approaches. Values of conductivities are in W/m·K.

Model	Isotropic		Anisotropic		Fractural	
**Parameters**	λ	8.0	$\lambda_{1}$	12.1	$\lambda_{1}^{\mathrm{shell}}$	17.9
			$\lambda_{2}$	3.5	$\lambda_{3}^{\mathrm{shell}}$	2.1
					$\lambda_{3}^{\mathrm{core}}$	4.2
**Objective value**	4.36		1.14		1.01	

## Data Availability

Not applicable.

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
