# Peer review of "Numerical Computation of Anisotropic Thermal Conductivity in Injection Molded Polymer Heat Sink Filled with Graphite Flakes"

_polymers, 2022, doi:10.3390/polym14163284_

Round 1
Reviewer 1 Report
1. The novelty of the paper should be explained more clearly.
2. There are some apparent errors, like ”Error! Reference source not found.”
3. Line 150, The directions of flows a,b, and c are unclear.
4. Line 173, there is a mistake in the formula (3).
5. Line 213, what’s the definition of NT?
6. Line 257, what is the definition and value of hc?
7. Line 269, the sensitivity analysis should be described more specifically.
Author Response
Our response to comments of Reviewer 1.
Point 1: The novelty of the paper should be explained more clearly.
Response 1: The novelty lies in modelling the material with a "planar" form of graphite as a filler. We have not come across such research in our literature survey, all complex models are concerned with graphite fibers (supported with injection molding simulations). Moreover, we could only find one reference that would match the internal structure of our material (reference 29 - Grundler). The fractural model showed the potential for further research and a way to model composites with graphite flakes. We believe that these statements/ideas are found in the introduction and conclusion, therefore, we have not added anything to the text.
Point 2: There are some apparent errors, like ”Error! Reference source not found.”
Response 2: We have not encountered this on our side, so we don’t know the origin of this problem. However, we have provided a version of the article where cross references have been replaced by plain text.
Point 3: Line 150, The directions of flows a,b, and c are unclear.
Response 3: We think that it is explained clearly. The letters a, b and c refer to the thermal conductivity measurement configurations. Here is the snippet from the manuscript:
„The measurement was performed in three different configurations:
- whole specimen, perpendicular to the flow (in the Figure 4 depicted as a) – overall through-plane conductivity,
- grinded specimen (the shells were grinded, so that the core itself could be measured), perpendicular to the flow (b) – through-plane conductivity in the core,
- grinded specimen, parallel to the flow (c) – in-plane conductivity in the core.“
We think that the summaries in bold text fully describe the relevant information using common thermal conductivity terminology.
Point 4: Line 173, there is a mistake in the formula (3).
Response 4: It has been corrected.
Point 5: Line 213, what’s the definition of NT?
Response 5: It is described in the Nomenclature: NT = Number of observed temperatures (accounted for in the objective function).
Point 6: Line 257, what is the definition and value of hc?
Response 6: It is defined in the Nomenclature: hc = Contact conductance, describing thermal contact between separate surfaces (boundaries). A table with input parameters to the numerical model is added (Table 1). The value of hc can be found in the table.
Point 7: Line 269, the sensitivity analysis should be described more specifically.
Response 7: The paragraph in question has been slightly changed. We do not think it is necessary to go into all the details, as meshing can be a very subjective matter and this particular model solves only a stationary heat conduction problem.
Reviewer 2 Report
I think the article is a value contribution to the field. I found it very explicative and easy to follow except for the problem with the cross references.
Minor comments
I miss a better description of the fabrication of the fin. There is not mention of the amount of graphite used (%wt or mg/ml) by this compound, and i didn't find any reference related to the product.
There is a typo in equation 3, ld = Ld/Rd,
I also miss a good thermal picture of the system and its comparison with the simulated fin. There is their previous work, but one extra here can give you a better visual comparison between the simulation and measurements.
I didn't find the value of the emissivity.
Author Response
Our response to comments of Reviewer 2.
Point 1: I miss a better description of the fabrication of the fin. There is not mention of the amount of graphite used (%wt or mg/ml) by this compound, and i didn't find any reference related to the product.
Response 1: We have added the technical designation of the material in the chapter Heat Sink Prototype. More text describing the manufactory process was added. Figure 1 now includes a photo of the mold that was used in production. Regarding the amount of graphite used, it is given by the commercial product, we have not investigated that for our research.
Point 2: There is a typo in equation 3, ld = Ld/Rd.
Response 2: It has been corrected.
Point 3: I also miss a good thermal picture of the system and its comparison with the simulated fin. There is their previous work, but one extra here can give you a better visual comparison between the simulation and measurements.
Response 3: We have added a new figure (now figure 13) with side by side comparison. The thermal image shown is one of many that were used to obtain the average temperature at the top of the fins.
Point 4: I didn't find the value of the emissivity.
Response 4: A table with input parameters to the numerical model has been added (Table 1). The value of the emissivity can be found in the table.
Reviewer 3 Report
The authors performed numerical computation on an injection molded polymer heat sink with graphite flake mixed as conduction filler. Based on three assumptions, the authors provided three cases/models to solve the heat conduction problem, including homogeneous conduction, anisotropic conduction, and the so-called fraction anisotropic conduction. For different cases, they treated the model at different microscopic level details. Especially the last case, which obtaining information from the internal structure of the injection molded part, by capturing the orientation of the graphite flakes. They hope to get better results by considering more detailed structure, and the results show this trend, but the quality improvement is somehow not quite significant. They also mentioned that the fractional method could be more useful for complex part, which is sounded reasonable. This work presented a comparative case study for applying the three assumptions to the same heat sink samples and provide useful suggestions on how to optimize parameters to get more accurate predictions. I would like to recommend this paper to be published on Polymers after authors address the following major issues.
(1) Reference labels are massively missing, and some error codes are found at the reference citation location. Please correct them.
(2) When referring an equation in the text, please use the following format: “Eq. 3 shows …”, instead of only using a number to represent equation number.
(3) From the core-shell assumption, it seems like the determination of the boundary of the core-shell interface is very important to get the correct conductivity of each part. I cannot find how the core-shell structure was defined and how to determine which part belong to core and which part belong to shell from the internal image of the heat sink. Please clarify it. Please also add discussion how different choices affect the final computation results and give some suggestions on how to optimize it to minimize the error.
(4) I hope authors adding one additional table summarizing all input parameters used to solve the three models before showing the final results. Doing these will help other people reproduce your computational results.
(5) When considering the boundary condition of the air flow for the surface of between the fins, a constant air temperature 20 degree C is used. Does it match the experiments? It looks like the air can be heated up when flow through the fins, could the different air flow boundary condition affect the final results? Please add some discussions to the main text to clarify this.
(6) It looks like the traditional anisotropic model shows good-enough accuracy, do you think the proposed structural based fractural method worth the time/effort for improving the accuracy of the prediction?
Author Response
Our response to comments of Reviewer 3.
Point 1: Reference labels are massively missing, and some error codes are found at the reference citation location. Please correct them.
Response 1: We have not encountered this on our side, so we don’t know the origin of this problem. However, we have provided a version of the article where cross references have been replaced by plain text.
Point 2: When referring an equation in the text, please use the following format: “Eq. 3 shows …”, instead of only using a number to represent equation number.
Response 2: Equation references have been modified.
Point 3: From the core-shell assumption, it seems like the determination of the boundary of the core-shell interface is very important to get the correct conductivity of each part. I cannot find how the core-shell structure was defined and how to determine which part belong to core and which part belong to shell from the internal image of the heat sink. Please clarify it. Please also add discussion how different choices affect the final computation results and give some suggestions on how to optimize it to minimize the error.
Response 3: We think that the core-shell definition is clearly stated in the text. However, we reformatted the information into two bullets to highlight how we define the structures based on the flakes’ orientation. It is as follows:
- The flakes in the core are arranged in a parabolic shape perpendicular to the direction of material flow during fabrication.
- Conversely, in the shell they form dense parallel lines in the flow direction.
Moreover, a paragraph has been added after Figure 7 describing how we determined the core/shell dimensions.
Regarding the discussion, it is important to look not only at the temperature error versus experiment, but also at the resulting conductivities to see if they are within expected limits. Before determining the core/shell dimensions (using only pure estimate without looking at the internal structure), our conductivities were different from the expected values. Only after we determined the dimensions of the structure from microscopic photographs (approximately since the core-shell interface is complex), thus removing the unknown parameter, did we obtain conductivity values within acceptable limits.
Point 4: I hope authors adding one additional table summarizing all input parameters used to solve the three models before showing the final results. Doing these will help other people reproduce your computational results.
Response 4: A table with input parameters to the numerical model has been added (Table 1) at the end of chapter 3 Numerical model.
Point 5: When considering the boundary condition of the air flow for the surface of between the fins, a constant air temperature 20 degree C is used. Does it match the experiments? It looks like the air can be heated up when flow through the fins, could the different air flow boundary condition affect the final results? Please add some discussions to the main text to clarify this.
Response 5: A paragraph has been added. It describes using a separate CFD simulation to obtain estimates of the heat transfer coefficient that were applied as temperature dependent interpolation functions in the boundary condition for natural convection. The heat transfer coefficient was related to the air temperature of 20 degree C. That is how we treated the complexity of air flow.
Point 6: It looks like the traditional anisotropic model shows good-enough accuracy, do you think the proposed structural based fractural method worth the time/effort for improving the accuracy of the prediction?
Response 6: We think it is worth putting effort into creating a structural model. If only for the reason that the conductivities obtained for shell/core can be applied to a forward problem on a different geometry (if we are talking about desire to predict the thermal behavior of a product). Although the traditional anisotropic model achieved comparable results in this case, the conductivity values do not have the same logical basis as shell/core and we think they are not transferable to other geometries. If the height of the heat sink base was doubled, we wouldn't be able to tell if we can really use the conductivities from the traditional anisotropic model. With knowledge of the core and shell dimensions, we can more confidently use the conductivities from the fractural method.
If we considered a more complex geometry, the temperature differences could be larger and the temperature near the heat source would not match, which is crucial in design.
Since we have not encountered this in literature, we think that our research is sort of pioneering, therefore, we currently do not have data to support our assumptions.
Round 2
Reviewer 3 Report
All points are addressed properly, accept as is.